

# Systematic review of residual toxicity studies of pesticides to bees and veracity of guidance on pesticide labels

Leah Swanson[1], Andony Melathopoulos[2] and Matthew Bucy[3]

[1] Oregon State University, Corvallis, OR, United States of America
[2] Department of Horticulture, Oregon State University, Corvallis, OR, United States of America
[3] Oregon Department of Agriculture, Salem, OR, United States of America

## ABSTRACT

Residues of pesticides on crops can result in mortality to foraging bees. Pesticide applicators in the U.S. encounter a statement on pesticide labels, which coarsely indicate which products dissipate over the course of an evening. There is reason to suspect that these statements may not align with residual toxicity data, given previous findings. Without a complete database of residual toxicity estimates; however, it is not possible to determine whether the residual toxicity components of statements on pesticide labels similarly diverge from published studies. We compiled 50 studies on residual toxicity trials with formulated pesticides and calculated the residual time to 25% mortality ($RT_{25}$) of each assay for three different bee species (*Apis mellifera, Nomia melanderi,* and *Megachile rotundata*). Our findings were compared to a U.S. Environmental Protection Agency (EPA) published database of $RT_{25}$ values. Of the $RT_{25}$ values that we could compare, we found that over 90% of the values support a similar conclusion to the EPA. Next, we compared our values and the EPA's values to the statements on 155 EPA registered pesticide product labels. Of these labels, a little less than a third presented their residual toxicity in a manner inconsistent with their calculated $RT_{25}$ and current EPA labeling guidelines. Moreover, over a third of labels contained an active ingredient which was neither listed under the EPA's $RT_{25}$ database nor had a published study to estimate this value. We provide the first evidence that many pesticide labels may convey residual toxicity information to applicators that is not correct and could lead to bees being exposed to toxic residues on plants.

# INTRODUCTION

Pesticides can have negative impacts on individual bees and bee colonies when toxic products are applied to blooming plants that are bee attractive (*Botías et al., 2017*; *Chauzat et al., 2010*; *Kiljanek et al., 2017*; *Tosi et al., 2018*; *Graham et al., 2021*). Bees can become exposed to pesticide residues when foraging on pesticide-treated plants, which can result in mortality if the residues are at levels that are acutely toxic to them. Mortality, however, may be lessened if the pesticide is applied in the evening, when bees are not foraging. Theoretically, this allows for an interval over which the pesticide can dissipate on the

Corresponding author
Andony Melathopoulos, Andony.Melathopoulos@oregonstate.edu

plant sufficiently to become relatively non-toxic to bees when they resume foraging the next day (*Johansen et al., 1983a*; *Johansen et al., 1983b*). Evening pesticide applications as a way to mitigate exposure, however, are predicated on the assumption that the residues of the pesticides will weather sufficiently before bees resume foraging the next morning and come into contact with treated leaves and flowers (*Barmaz, Potts & Vighi, 2010*; *Fischer & Moriarty, 2011*; *The Honey Bee Health Coalition, 2019*; *Smodiš Škerl et al., 2009*).

The rate at which acute toxicity of pesticides to bees dissipates from plant surfaces is known as the pesticide's residual time. Pesticide registrants in the U.S. are required to estimate the residual time for all formulated pesticides that contain one or more active ingredients that is acutely toxic to bees (*i.e.,* acute contact toxicity lethal dose to 50% of the honey bees ($LD_{50}$) is less than 11 micrograms of pesticide per bee) and the use pattern indicates that bees are likely to be exposed (40 CFR 158.630(d)). The U.S. Environmental Protection Agency (EPA) provides guidance for registrants on how to conduct a trial to estimate residual time (*United States Environmental Protection Agency, 2012a*). These trials involve spraying a field crop (typically alfalfa) with a pesticide, allowing residues to weather for set intervals, then harvesting plant material and placing it in a cage with honey bees (*Apis mellifera*). The bees are free to walk over the plant material for a set period of time (typically 24 h), after which the number of dead bees is counted. Residual time is expressed as the weathering interval after which the mortality of bees contacting the foliage reaches 25% mortality (referred to as the residual time to 25% mortality or $RT_{25}$). The basic pattern of these trials pre-dates EPA guidelines and have been used by toxicologists since the 1960s (*e.g.,* *Wiese, 1962*). Notably, this approach does not take into account the time taken for a systemic pesticide to no longer be present in nectar and pollen, which is addressed elsewhere in the EPA's risk assessment for bees (*United States Environmental Protection Agency, 2016*)

A key threshold residual time identified by the EPA is known as extended residual toxicity. A pesticide with extended residual toxicity is one that cannot be applied safely in the evening as residues would cause more than 25% mortality of bees in a cage assay. Although the residual time threshold is not specified in the EPA's Label Review Manual (*United States Environmental Protection Agency, 2012b*); elsewhere, the EPA indicates that a pesticide with extended residual toxicity has $RT_{25}$ >8 h (*Office of Chemical Safety and Pollution Prevention, 2012*). Typically, pesticide labels in the U.S. only indicate whether pesticides that are acutely toxic to bees have extended residual toxicity or not and generally do not list $RT_{25}$ values (*Office of Chemical Safety and Pollution Prevention, 2012*).

$RT_{25}$ is an important tool in determining how to best mitigate the risk of bee exposure to pesticides residues. The importance of $RT_{25}$ estimates for pesticide applicators when selecting and applying a pesticide is evinced by state Cooperative Extension publications that list $RT_{25}$ values from published studies (*e.g.,* *Hooven, Sagili & Johansen, 2013*). Furthermore, $RT_{25}$ estimates are used by the EPA in order to characterize the hazards and risks of pesticides to pollinating insects. The EPA requires that a product's residual toxicity to bees be communicated on the product label in a way that is reflective of the $RT_{25}$ value. The EPA has produced guidance for their reviewers and pesticide registrants on the language they will typically suggest for different $RT_{25}$ values (*United States Environmental Protection*

*Agency, 2012b*). This information will typically be available in the Environmental Hazards section of the label, but it is not federally enforceable and is used as an informational tool for pesticide applicators (*United States Environmental Protection Agency, 2012b*). However, pesticide labels rarely state the $RT_{25}$ value, so this information is not readily accessible to pesticide applicators, crop advisors or extension educators. There remains a demand for better guidance on the dissipation rates of bee toxic products under field conditions.

Notably, more recent EPA guidance (*United States Protection Agency, 2017*) provides more specific mitigation language around the extended residual toxicity threshold for the safe application of pesticides during bee pollination. These new guidelines provide federally enforceable specific use instructions for residual toxicity stating that if extended residual toxicity (residues persisting for greater than 8 h *i.e.*, extended residual toxicity) is not present for a pesticide it can be applied 2 h before sunset when pollinators are least active (*United States Protection Agency, 2017*). Pesticide registrants have begun adopting this guidance, one example is Harvanta 50SL (Summit Agro™, Durham, NC, EPA registration number 71512-26-88783), which states for fruiting vegetables (Crop Group 8-10) "foliar application of this product is prohibited to a crop from onset of flowering until flowering is complete unless the application is being made in the time period between 2 h prior to sunset until sunrise". While this shows that some labels have been written in accordance with this new policy, many pesticide labels still follow pre-2017 guidance in communicating residual toxicity to bees (*e.g.*, Product Dursban 50W, EPA Registration Number 62719-72; Product Merit 2F, EPA Registration Number 432-1312, which states: "do not apply this product or allow it to drift to blooming crops or weeds if bees are visiting the treatment area"). In addition to the 2017 guidance, the EPA released a public summary of $RT_{25}$ estimates compiled from registrant-submitted data to the public (*United States Environmental Protection Agency, 2014*). Notably, the summary only included studies that have "undergone quality assurance reviews to ensure that the data are scientifically sound", and, in turn, is missing several widely used active ingredients (*e.g.*, bifenthrin). Regardless, the omissions pose a challenge to researchers looking to compare pesticide label language on residual toxicity to $RT_{25}$ values.

There is a need to investigate pesticide label language against studies that characterize environmental risks. *Bucy & Melathopoulos (2020)*, for example, found that roughly 32% of pesticide labels analyzed had at least one error in the communication of acute toxicity to bees, or the adverse effects caused after a short exposure time to an active ingredient (OCSPP 850.3000). These authors, however, were unable to do a similar analysis with residual toxicity statements because of the absence of a comprehensive database of $RT_{25}$ values.

Our objective was to provide the first analysis of pesticide label statements communicating residual toxicity to bees in comparison to actual $RT_{25}$ values. The overall reason for doing this is to ensure that residual toxicity information is correctly communicated to pesticide applicators on labels. We approached the challenges experienced by *Bucy & Melathopoulos (2020)* by creating a database of $RT_{25}$ values to compare to pesticide label statements. Our approach to creating a database was to assemble all published residual toxicity studies and characterize variability in methodologies used to

assess residual toxicity. We then conducted a systematic review to calculate $RT_{25}$ estimates for each pesticide and validated these estimates against values published by EPA (*United States Environmental Protection Agency, 2014*). We used the validated database to analyze the residual toxicity statement on pesticide labels and to determine how $RT_{25}$ estimates vary by the rate of pesticide used, the formulation of the pesticide, and bee species.

## MATERIALS & METHODS

### Selection of studies

Portions of this manuscript were previously published as part of a preprint (https://www.biorxiv.org/content/10.1101/2023.06.05.543089v1.full). We located putative residual toxicity studies using Web of Science with the search term "residual toxicity" as well as the names of bee taxa commonly used in residual toxicity assays and currently listed in EPA's $RT_{25}$ database: "*Apis*", "*Nomia*" and "*Megachile*". This search returned a total of 130 studies. Next, we located residual toxicity studies on the alfalfa leafcutting bee (*Megachile rotundata*) from proceedings of the Western Alfalfa Seed Growers Association (2004-2017), resulting in an additional 17 studies. We also evaluated a series of Bee Research Investigation and Integrated Pest and Pollinator Investigation reports released by Washington State University amounting to 28 reports. Finally, we obtained 8 studies directly from Bayer CropSciences. In sum, we evaluated 183 residual toxicity studies. Databases were last searched in 2022. We narrowed these studies down to 50 by only including studies that met any of the following criteria: (1) the study was a primary source of data (*e.g.*, not a review paper); (2) bees were exposed to the pesticide applied to on plant material (*e.g.*, no studies where pesticide was applied to filter paper); and/or, (3) the study focused on residual toxicity of pesticides applied to plots of crop plants and involved harvesting plant tissue for caged bees. These criteria were designed to ensure that we only included studies whose residual toxicity methodology broadly followed those of the *United States Environmental Protection Agency (2016)*. We removed two additional studies because the author indicated that it was likely that some live bees in the assay were mistakenly counted as dead (*Johansen, Kious & Mayer, 1981*) and because the actual active ingredient of the product used was not specified (*Walsh et al., 2011*).

### Evaluation of studies

This analysis consisted of residual toxicity studies where a pesticide was foliar applied onto a specific crop, and the plant material (*e.g.*, foliage) was harvested at varying time intervals after application. The plant material was then collected and placed in cages with adult bees to contact for 24 h or longer. Residual toxicity was calculated from variation in bee mortality for bees exposed to plant material harvested at different intervals of weathering. We defined each time interval that plant material was collected at as one trial. Although studies were selected based on their broad adherence with the methodology developed by (*United States Environmental Protection Agency, 2016*), they varied across several test parameters. We categorized the variance from EPA methodology across four key study parameters (Table 1).

**Table 1  Descriptions of study design elements examined during the meta-analysis.**

| Study design element | Variables examined | USEPA guidelines |
|---|---|---|
| Bees[a] | 1. Caste/sex | 1. Female worker bees |
| | 2. Age | 2. Young |
| Plant Materials[b] | 1. Crop | 1. Alfalfa |
| | 2. Plant part | 2. Foliage |
| Exposure[c] | 1. Number of bees per cage | 1. 25 per cage |
| | 2. Plant part weight mass | 2. 15 grams |
| | 3. Duration of mortality observation | 3. Greater than or equal to 24 h |
| Environmental Conditions[d] | 1. Temperature | 1. 25 to 35 degrees Celsius |
| | 2. Syrup provided | 2. Yes |
| | 3. Syrup concentration | 3. 50:50 weight to volume |

**Notes.**

[a] Consists of the caste, sex, age, and source of bees placed in the cage during the residual toxicity trail. Caste = either worker, drone, or queen. Sex = either male or female. Age of bees = how old the bees (in days) generally were.

[b] Pertains to the materials used during the residual toxicity trials. Crop = the type of crop the product was sprayed on. Plant part = the part of the plan placed in the cages with the bees.

[c] How the bees were exposed to the pesticide. Number of bees per cage = the number of bees placed in each cage during the residual toxicity trial. Plant part weight mass = the weight mass of the plant part placed in the cage during the trial. Duration of mortality observation = how long in hours the bees were observed for mortality after being exposed.

[d] The environmental conditions that the bees were held at during the residual toxicity trial. Temperature = the average temperature the bees were incubated at during the trial. Syrup provided = if syrup was provided during the observation period. Syrup concentration = the concentration of the syrup in terms of water to sucrose.

We used the following approaches to standardize methodologies across studies. The EPA uses the word "young" to describe the optimal age of bees for residual toxicity trials. We interpreted "young" to mean newly emerged adult (eclosed) bees that were less than 1 day of age (*Winston, 1991*). Furthermore, when a study reported a range for a parameter, such as for number of bees per cage or temperature, we used the average calculated from the low and high points of the range. In reference to the diet that the bees were fed during the assay, one study reported the syrup concentration as 91:1 (wt:wt) which we assumed was 1:1 (*Mayer, 2001*).

We evaluated whether parameters in studies aligned with EPA recommendations, by counting each testing parameter as described in 'Evaluation of studies'. We noted whether studies had test parameters that corresponded to those recommended by the EPA or if there was not enough information to determine correspondence.

## Calculation of RT$_{25}$ values

Very few studies report RT$_{25}$ values, instead reported the number of bees alive or dead at different time intervals. In order to compare the values in these studies to EPA's RT$_{25}$ values we used the mortality data from different time intervals to estimate RT$_{25}$. Trials within studies were compiled by active ingredient, the formulation of the pesticide product (emulsifiable concentrate, wettable powder, *etc.*), application rate, the species of bee used in the cage assay and the duration residues were allowed to weather. We removed any trials with only a single weathering period because these could not be used to calculate residual time. We also removed studies if they did not specify application rates, or percentage mortality and if mixtures of active ingredients were used. The EPA includes both *M.*
*rotundata* and *N. melanderi* as well as *A. mellifera* in their published $RT_{25}$ values but not *Bombus*. Consequently, we also removed *Bombus* trials from this analysis since comparison to the EPA would have been impossible.

We used R statistical software (v4.1.1; *R Core Team, 2021*) along with the package Tidyverse (*Wickham et al., 2019*) to calculate the $RT_{25}$ values using regression models where time was the independent variable and percent mortality the dependent variable. We checked for overdispersion in all assays. If the data was not over dispersed, we then calculated the $RT_{25}$ values through a binomial logistic regression. If the data was over dispersed, we calculated the $RT_{25}$ values using a quasibinomial logistic regression.

## Comparison of $RT_{25}$ values

We validated the database created from calculated $RT_{25}$ values ('Calculation of $RT_{25}$ values') by comparing residual times for each active ingredient by application rate, formulation of the product (*i.e.,* emulsifiable concentrate, wettable powder, etc.), and species of bee in the database published by the EPA (*United States Environmental Protection Agency, 2014*). Instead of comparing the $RT_{25}$ estimates themselves, we compared how each database would categorize a pesticide as having extended residual toxicity or not. For example, if our calculated $RT_{25}$ value for a pesticide was less than 6 h (*i.e.*, no extended residual toxicity) and EPA indicated the $RT_{25}$ value was greater than 12 h (*i.e.*, extended residual toxicity), we deemed the two as sufficiently different. Furthermore, we assumed EPA database estimates were accurate. If extended residual toxicity determinations matched those of pesticides from our systematic review, this would mean that we could rely on $RT_{25}$ estimates for active ingredients that did not appear in the EPA database. In contrast, if there were substantial misalignment among extended residual toxicity determinations between our calculated and USEPA $RT_{25}$ estimates, we would conclude that our calculation methods significantly differed from EPA's and our estimates would need to be reevaluated.

We compared $RT_{25}$ values for bee species across active ingredient, formulation, and application rate. *M. rotundata* and *N. melanderi* $RT_{25}$ values were compared to *A. mellifera* $RT_{25}$ values since, currently, the EPA generally only requires registrants to conduct residual toxicity assays for *A. mellifera* when applying for product registration. In doing so, we were able to determine whether Environmental Hazards language reflects the $RT_{25}$ estimates of *M. rotundata* and *N. melanderi*.

## Label language analysis

We created a composite database of $RT_{25}$ values from the EPA (*United States Environmental Protection Agency, 2014*) supplemented with calculated values based on the findings of 'Comparison of $RT_{25}$ values'. To determine if $RT_{25}$ values correspond with residual toxicity statements under the Environmental Hazard section of pesticide labels, we used an existing database of residual toxicity statements on pesticides labels developed by *Bucy & Melathopoulos (2020)* and compared it to $RT_{25}$ values in our composite $RT_{25}$ database. The database consisted of 232 labels obtained from products that were used: on 12 Oregon crops around bloom (alfalfa seed, apple, blueberry, carrot seed, cherry, clover seed, cranberry, meadowfoam, pear, pumpkin/squash, radish seed, and watermelon) and

California almonds, in Oregon Christmas tree fields during peak times of honey bee activity, to control mosquitos any time of the year, and as garden products available throughout the year to Oregon consumers. We excluded labels from this analysis if: (1) the Environmental Hazards indicated the product was not 'toxic' or not 'highly toxic' to bees. This would mean that the active ingredient has an $LD_{50}$ for bees of greater than 11μg/bee, in which case the EPA would not have required that the registrant assess the residual toxicity of that product and/or (2) the product was unlikely to result in exposure to bees (*e.g.*, granular formulations). We only used *A. mellifera* $RT_{25}$ values in this analysis since residual toxicity language on pesticide labels is specific to this species (*United States Environmental Protection Agency, 2012b*). Similar to *Bucy & Melathopoulos (2020)*, We interpreted pesticides with short residual toxicity ($RT_{25}$ <8 h) as corresponding to the statement: "Do not apply…while bees…**are actively foraging** the treatment area" and those with extended residual toxicity ($RT_{25}$ >8 h) if accompanied with the statement: "Do not apply…if bees…**are foraging** the treatment area" found in EPA label language guidance (Fig. 1; *United States Environmental Protection Agency, 2012b*). We compared the $RT_{25}$ values to the residual toxicity statements on labels for the same formulation (*e.g.*, emulsifiable concentrate, wettable powder, etc.) of the same active ingredients between the calculated $RT_{25}$ values and the pesticide.

The following assumptions were made regarding the interpretation of slight variation from EPA guidance when reviewing labels. "Bees are least active" was interpreted as "Do not apply…while bees…**are actively foraging** the treatment area" and "Bees may forage" was interpreted as "Do not apply…if bees…**are foraging** the treatment area". If no label language associated with $RT_{25}$ values was present on the label, the data from that label was included as "N/A" in analysis. If no acute toxicity language was present on the label suggesting a $LD_{50}$ of greater than 11 μg/bee, we excluded the active ingredient from our analysis. If an active ingredient had an $LD_{50}$ greater than 11 μg/bee, EPA would not require an acute or residual toxicity statement on the product label.

We determined misalignment between the Environmental Hazards and $RT_{25}$ estimates based on the extended residual toxicity threshold (see 'Comparison of $RT_{25}$ values'). For example, if a label suggested an $RT_{25}$ value less than 8 h but the database indicated an $RT_{25}$ estimate that was greater than 8 h, we deemed the label language as not aligning. If a label was not aligning, we further categorized the labels as either having language that was interpreted as a longer $RT_{25}$ value than we had calculated or as having language that was interpreted as a shorter $RT_{25}$ value than we had calculated. Many labels had language that corresponded with $RT_{25}$ values but neither EPA nor the literature examined during the systematic review had information on the active ingredient in the product or a similar formulation to compare the two. These labels were included in the analysis as labels that had "$RT_{25}$ values missing".

# RESULTS

## Methodology of residual toxicity trials

Almost three quarters of the studies (70%) analyzed used EPA's recommended leaf foliage as the treated plant material placed in cages during the residual toxicity trials, with around
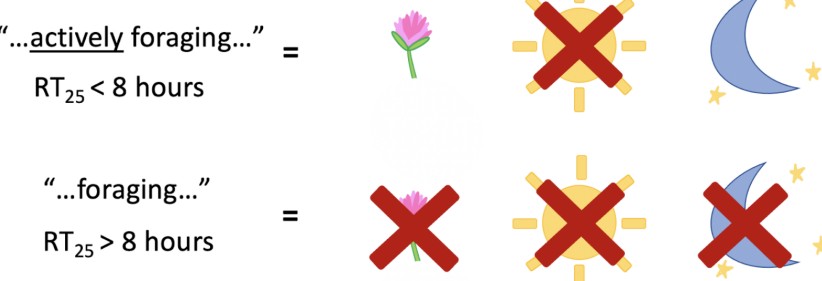

[1] Pesticide products used for the calculation of RT$_{25}$ values for each active ingredient can be found in the Residual Toxicity Data supplementary material.

**Figure 1** **Simplified overview of how residual toxicity corresponds with language found on pesticide labels and the application procedures of a pesticide.** On the left, the language found on the pesticide label and what RT$_{25}$ values each would correspond with. On the right, whether the pesticide could be applied during bloom, during the day, or during the night.

a quarter of the studies using other materials such as flowers. In all studies, bees stocked in cages with treated plant material were fed sucrose syrup *ad libitum*. A majority of studies (69%) aligned with the EPA recommendation for a 50% (wt:wt) sucrose to water solution (Fig. 2). The temperature at which bees were incubated during the residual toxicity test varied greatly among the studies. Most studies incubated bees outside the temperature range of 25−35 °C as recommended by the EPA (Fig. 2), tending to incubate at cooler temperatures. The crop used in studies was evenly distributed between the EPA recommendation of alfalfa and other crops. The studies that did not use alfalfa used, in descending order of frequency, cotton, white clover, strawberry, and sunflower. About half of the studies (48%) reported that there were 25 bees placed in the cage for each residual toxicity trial as recommended by the EPA, with remaining studies ranging from 10 to 106 bees per cage. On average, trials using *A. mellifera* had more bees (56) per cage compared to *M. rotundata* (24 bees per cage) and *N. melanderi* (20 bees per cage). The age of the bees used during the residual toxicity trial mostly deviated across studies with almost half of studies (46%) using an older age of bees (>1 day old) than recommended by the EPA.

### RT$_{25}$ calculations and comparisons

We calculated RT$_{25}$ values from 135 of 490 trials in the 50 studies that were reviewed. We were unable to calculate RT$_{25}$ values for the other 355 trial because published mortality percentage values were either above 25% for all time periods reported or were below 25% for the duration of the assay. In these cases, we indicated the RT$_{25}$ value as greater than the longest reported period or less than the shortest reported period respectively.

When comparing RT$_{25}$ values across different formulations, there were six cases (1.4%) where different formulations with the same rate of the same active ingredient resulted in different RT$_{25}$ values (Table 2). These cases were acephate, dimeothate, fipronil, formetanate hydrochloride, naled and trichlorofon. For the same formulation of the same active ingredient with different application rates, 20 active ingredients had different RT$_{25}$ values with higher application rates having typically longer RT$_{25}$ values. When comparing across species, there were 21 cases (5%) where different species (*A. mellifera, M.*

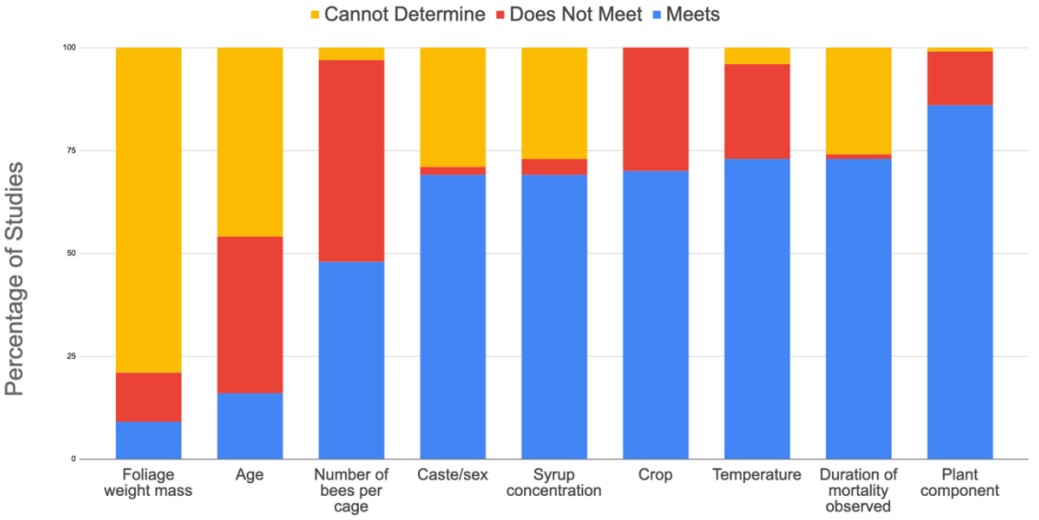

**Figure 2** Comparison of methodological parameters of residual toxicity studies ($n = 48$) with percentage of studies that meet EPA residual toxicity criteria (*United States Environmental Protection Agency, 2012a*), do not meet, and cannot determine.

*rotundata,* and *N. melanderi*) resulted in different calculated $RT_{25}$ values even though they were exposed to residues of an active ingredient that was applied as the same formulation, at the same application rate, and allowed to weather for the same amount of time. The most variation in $RT_{25}$ times was seen in active ingredients with the formulation of emulsifiable concentrate with *M. rotundata* consistently having longer $RT_{25}$ times (9 cases) compared to *A. mellifera* (Fig. 3). Finally, there were 11 cases (2.5%) where the same active ingredient at the same formulation applied at the same application rate tested on the same species resulted in contradicting $RT_{25}$ values.

Overall, calculated $RT_{25}$ values from studies matched the extended residual toxicity threshold reported by EPA with a single active ingredient that was not aligned. Notably, this single case, disulfoton emulsifiable concentrate applied at a rate of 1 pound of active ingredient per acre, was close to the extended residual toxicity threshold with a calculated $RT_{25}$ value of 8.86 h and a published EPA $RT_{25}$ value of 5.5 h. From our database, we were able to calculate $RT_{25}$ values for an additional 29 active ingredients that were not present in the EPA's published database (Fig. 4; *United States Environmental Protection Agency, 2014*).

## Label language comparisons

Based on the high level of the extended residual toxicity threshold agreement between EPA's published values (*United States Environmental Protection Agency, 2014*) and the database generated through our systematic review, we supplemented the EPA database with residual toxicity values for pesticides not previously included. Notably, even after supplementing the EPA database, we were still unable to compare the residual toxicity language on the Environmental Hazards section of labels for one third of pesticides due to a lack of data. Of the remaining labels, a third had residual toxicity warnings that corresponded to $RT_{25}$ values and 27% failed to have any residual toxicity warning despite being toxic to bees. Of

the cases where the $RT_{25}$ values did not correspond to residual toxicity statements, 17% of labels had a statement indicating that the product would remain toxic longer than the $RT_{25}$ value. The other 10% had a statement indicating the product would remain toxic shorter than the $RT_{25}$ value (Fig. 5).

## DISCUSSION

We have developed the most comprehensive database available for $RT_{25}$ values, which greatly expands the publicly accessible values initially published by the EPA in 2014. Using published studies, we were able to add 29 active ingredients in addition to the EPA's 70 active ingredient values. We demonstrated that while test methodologies varied among published studies, they are nonetheless consistent in determining whether a pesticide had extended residual toxicity or not, suggesting that variation in methodologies or the environmental conditions under which the tests were conducted did not result in substantively different conclusions on whether or not an applicator could apply the product at bloom in the evening. Despite our efforts to expand on EPA's existing $RT_{25}$ database, we found that there remains a paucity of residual toxicity studies. A third of labels in our database did not have corresponding $RT_{25}$ values in either the $RT_{25}$ values calculated as part of our study nor the $RT_{25}$ value estimates published by EPA. We took more variables such as formulation of the active ingredient into account than may have been necessary when calculating $RT_{25}$ which may have contributed to the lack of comparable $RT_{25}$ values. For example, future studies may choose to not consider such variables as formulation when calculating $RT_{25}$ values to maximize the amount of studies usable for calculating each individual $RT_{25}$ value. Moreover, most studies included in our analysis were published in the 1990s and numerous new active ingredients have since been registered, which illustrates the extent to which researchers have not kept pace with the rate of pesticide product development. Despite these challenges, our systematic review expanded EPA's $RT_{25}$ database and was able to draw attention to widespread misalignment between $RT_{25}$ values and the Pollinator Insect Hazard Statement on pesticide labels that informs pesticide applicators of products with extended residual toxicity.

There was general agreement on whether an active ingredient had extended residual toxicity (*i.e.*, an $RT_{25}$ value >8h) between our systematic review and EPA's database. We found only one deviation across 57 comparable studies of the same active ingredient and application rate. This agreement is remarkable since key aspects of the test methodology were not standardized. Our systematic review $RT_{25}$ estimates were often within 1–3 h of those published by the EPA. For example, we calculated the $RT_{25}$ for chlorpyrifos formulated as an emulsifiable concentrate on *A. mellifera* as 17 h compared to the EPA database estimate of 16 h. However, the approach used in this analysis to compare in terms of the extended residual toxicity threshold instead of point estimates may have reduced the influence of methodological variation. Using the extended residual toxicity threshold, the $RT_{25}$ value for the pyrethroid insecticide fenpropathrin at 0.4 lbs ai/A for *A. mellifera* was determined to be greater than 8 h while the EPA reported the value as less than 336 h. Thus, we would deem these two values as the same, because they both support a conclusion

**Table 2  Compiled Calculated RT$_{25}$ database.**

| Active Ingredient[1] | Formulation | Rate (lb ai/A) | Calculated RT$_{25}$ Values | | | Test Species Scientific Name |
|---|---|---|---|---|---|---|
| | | | **Meta-Analysis** | **# of Studies** | **EPA Reported Value** | |
| Acephate | LS | 1 | >24 | 1 | | *Apis mellifera* |
| | | 0.5 | >24 | 1 | | *Apis mellifera* |
| | | | >24 | 1 | | *Megachile rotundata* |
| | SP | | >8 | 1 | | *Apis mellifera* |
| | | 1.29 | >8 | 1 | | *Megachile rotundata* |
| | | | >8 | 1 | | *Nomia melanderi* |
| | | | 7 or >72 | 3 | | *Apis mellifera* |
| | WP | 1 | 7.2 or >72 | 3 | | *Megachile rotundata* |
| | | | 7.78 or >72 | 3 | | *Apis mellifera* |
| Acetamiprid | WP | | <2 | 1 | | *Apis mellifera* |
| | | 0.05 | <2 | 1 | | *Megachile rotundata* |
| | | | <2 | 1 | | *Nomia melanderi* |
| | | | <2 | 1 | | *Apis mellifera* |
| | | 0.075 | <2 | 1 | | *Megachile rotundata* |
| | | | <2 | 1 | | *Nomia melanderi* |
| | | | <2 | 1 | | *Apis mellifera* |
| | | 0.1 | <2 | 1 | | *Megachile rotundata* |
| | | | <2 | 1 | | *Nomia melanderi* |
| | | | <2 | 1 | | *Apis mellifera* |
| | | 0.15 | <2 | 1 | | *Megachile rotundata* |
| | | | >8 | 1 | | *Nomia melanderi* |
| | | | <2 | 1 | | *Apis mellifera* |
| | | 0.3 | <2 | 1 | | *Megachile rotundata* |
| | | | >8 | 1 | | *Nomia melanderi* |
| Aldoxycarb | F | 3 | >8 | 1 | | *Apis mellifera* |
| Azamethiphos | WP | 0.5 | >144 | 1 | | *Apis mellifera* |
| | | 2 | >144 | 1 | | |

*(continued on next page)*

**Table 2** (*continued*)

| Active Ingredient[1] | Formulation | Rate (lb ai/A) | Calculated RT$_{25}$ Values | | | Test Species Scientific Name |
|---|---|---|---|---|---|---|
| | | | Meta-Analysis | # of Studies | EPA Reported Value | |
| Azinphos-methyl | EC | 1 | >8 | 1 | | *Apis mellifera* |
| | | | >8 | 1 | | *Megachile rotundata* |
| | | | >8 | 1 | | *Nomia melanderi* |
| | WP | 1 | >8 | 1 | | *Apis mellifera* |
| | | | >8 | 1 | | *Megachile rotundata* |
| | | | >8 | 1 | | *Nomia melanderi* |
| Bifenthrin | E | 0.032 | 19 | 1 | | *Apis mellifera* |
| | | 0.05 | >24 | 1 | | *Apis mellifera* |
| | EC | 0.0125 | 63 | 1 | | *Apis mellifera* |
| | | | >72 | 1 | | *Megachile rotundata* |
| | | 0.025 | >72 | 1 | | *Apis mellifera* |
| | | | >72 | 1 | | *Megachile rotundata* |
| | | 0.05 | >72 | 1 | | *Apis mellifera* |
| | | | >72 | 1 | | *Megachile rotundata* |
| | | 0.06 | 128 | 1 | | *Apis mellifera* |
| | | 0.1 | >72 | 1 | | *Apis mellifera* |
| | | | >72 | 1 | | *Megachile rotundata* |
| | ULV | 0.06 | 81.2 | 1 | | *Apis mellifera* |
| Carbaryl | F | 3 | >8 | 1 | | *Apis mellifera* |
| | | | >8 | 1 | | *Megachile rotundata* |
| | | | >8 | 1 | | *Nomia melanderi* |
| | | 0.25 | >42 | 1 | >42 | *Apis mellifera* |
| | | 0.5 | >42 | 1 | >42 | *Apis mellifera* |
| | WP | | >48 | 2 | >42 | *Apis mellifera* |
| | | 1 | >48 | 1 | | *Megachile rotundata* |
| | | | >48 | 1 | | *Nomia melanderi* |
| | | 2 | >42 | 1 | >42 | *Apis mellifera* |
| Carbofuran | F | 0.245 | >8 | 1 | | *Apis mellifera* |
| | | | >8 | 1 | | *Megachile rotundata* |
| | | | >8 | 1 | | *Nomia melanderi* |
| | | | >336 | 1 | | *Apis mellifera* |
| | | 1 | 288 | 1 | | *Megachile rotundata* |
| | | | >72 | 1 | | *Nomia melanderi* |

**Table 2** (*continued*)

| Active Ingre-dient[1] | Formulation | Rate (lb ai/A) | Calculated RT$_{25}$ Values | | | Test Species Scientific Name |
|---|---|---|---|---|---|---|
| | | | Meta-Analysis | # of Studies | EPA Reported Value | |
| Chlorpyrifos | E | 0.75 | >8 | 1 | | *Apis mellifera* |
| | | | >8 | 1 | | *Megachile rotundata* |
| | | | >8 | 1 | | *Nomia melanderi* |
| | | 1 | >12 | 1 | | *Apis mellifera* |
| | | 1.5 | >8 | 1 | | *Apis mellifera* |
| | | | >8 | 1 | | *Megachile rotundata* |
| | | | >8 | 1 | | *Nomia melanderi* |
| | EC | 0.025 | >8 | 1 | | *Apis mellifera* |
| | | 0.05 | >8 | 1 | | *Apis mellifera* |
| | | 0.1 | >8 | 1 | | *Apis mellifera* |
| | | 0.25 | 17 | 1 | 16 | *Apis mellifera* |
| | | | >24 | 1 | >24 | *Megachile rotundata* |
| | | | 20 | 1 | 19 | *Nomia melanderi* |
| | | 0.5 | 99 | 2 | >24 | *Apis mellifera* |
| | | | 140 | 2 | >24 | *Megachile rotundata* |
| | | | 66.8 | 2 | >24 | *Nomia melanderi* |
| | | 1 | 141 | 2 | >24 | *Apis mellifera* |
| | | | 161 | 2 | >24 | *Megachile rotundata* |
| | | | >120 | 2 | >24 | *Nomia melanderi* |
| Clofentezine | F | 0.25 | <2 | 1 | | *Apis mellifera* |
| | | | <2 | 1 | | *Megachile rotundata* |
| | | 0.5 | <2 | 1 | | *Apis mellifera* |
| | | | <2 | 1 | | *Megachile rotundata* |
| | | 1 | <2 | 1 | | *Apis mellifera* |
| | | | <2 | 1 | | *Megachile rotundata* |
| | | 2 | <2 | 1 | | *Apis mellifera* |
| | | | <2 | 1 | | *Megachile rotundata* |
| Colpyralid | EC | 0.05 | <2 | 1 | | *Apis mellifera* |
| | | | <2 | 1 | | *Megachile rotundata* |
| | | 0.1 | <2 | 2 | | *Apis mellifera* |
| | | | <2 | 1 | | *Megachile rotundata* |
| | | 0.2 | 5 | 1 | | *Megachile rotundata* |
| | EW | 0.05 | <2 | 1 | | *Apis mellifera* |
| | | | 3 | 1 | | *Megachile rotundata* |
| | | 0.1 | <2 | 3 | | *Apis mellifera* |
| | | | 44.6 | 2 | | *Megachile rotundata* |
| | | | <8 | 1 | | *Nomia melanderi* |
| | | 0.2 | <2 | 1 | | *Apis mellifera* |
| | | | >72 | 1 | | *Megachile rotundata* |

**Table 2** (*continued*)

| Active Ingredient[1] | Formulation | Rate (lb ai/A) | Calculated RT₂₅ Values | | | Test Species Scientific Name |
|---|---|---|---|---|---|---|
| | | | **Meta-Analysis** | **# of Studies** | **EPA Reported Value** | |
| Cyfluthrin | E | 0.025 | >24 | 1 | | *Apis mellifera* |
| | | | >24 | 1 | | *Megachile rotundata* |
| | | 0.05 | >24 | 1 | >240 | *Apis mellifera* |
| | | | >24 | 1 | | *Megachile rotundata* |
| Cyhalothrin | E | 0.05 | >8 | 1 | | *Apis mellifera* |
| | | | >8 | 1 | | *Megachile rotundata* |
| | | | >8 | 1 | | *Nomia melanderi* |
| | | 0.01 | <2 | 1 | | *Apis mellifera* |
| | | | >8 | 1 | | *Megachile rotundata* |
| | | | 2 | 1 | | *Nomia melanderi* |
| | | 0.015 | <2 | 1 | | *Apis mellifera* |
| | | | 8 | 1 | | *Megachile rotundata* |
| | | | 3.64 | 1 | | *Nomia melanderi* |
| | EC | 0.02 | <2 | 1 | | *Apis mellifera* |
| | | | >8 | 1 | | *Megachile rotundata* |
| | | | 3.24 | 1 | | *Nomia melanderi* |
| | | 0.025 | 4.28 | 1 | | *Apis mellifera* |
| | | | >8 | 1 | | *Megachile rotundata* |
| | | | 6.54 | 1 | | *Nomia melanderi* |
| | | 0.03 | >8 | 1 | | *Apis mellifera* |
| | | | >8 | 1 | | *Megachile rotundata* |
| | | | >8 | 1 | | *Nomia melanderi* |
| Cypermethrin | E | 0.05 | >8 | 1 | >96 | *Apis mellifera* |
| | | | >8 | 1 | | *Megachile rotundata* |
| | | | >8 | 1 | | *Nomia melanderi* |
| | | 0.1 | >24 | 1 | | *Apis mellifera* |
| | EC | 0.06 | >8 | 1 | | *Apis mellifera* |
| | | 0.09 | 313 | 1 | | *Megachile rotundata* |
| | | 0.14 | 197 | 1 | | *Apis mellifera* |
| | ULV | 0.09 | 63.8 | 1 | | *Apis mellifera* |
| Cyromazine | WP | 025 | <2 | 1 | | *Apis mellifera* |
| | | 0.3 | >8 | 1 | | *Megachile rotundata* |
| | | | >8 | 1 | | *Nomia melanderi* |
| Deltamethrin | EC | 0.02 | 4.95 | 1 | 5.2 | *Apis mellifera* |
| | | | <2 | 1 | | *Apis mellifera* |
| | | 0.2 | 4.09 | 1 | | *Megachile rotundata* |
| | | | <2 | 1 | | *Nomia melanderi* |

**Table 2** (*continued*)

| Active Ingredient[1] | Formulation | Rate (lb ai/A) | Calculated RT₂₅ Values | | | Test Species Scientific Name |
|---|---|---|---|---|---|---|
| | | | **Meta-Analysis** | **# of Studies** | **EPA Reported Value** | |
| Diazinon | EC | 0.05 | >24 | 1 | | *Apis mellifera* |
| | | | >8 | 1 | | *Megachile rotundata* |
| | | 0.75 | >8 | 1 | | *Apis mellifera* |
| | | | >8 | 1 | | *Megachile rotundata* |
| | | | >8 | 1 | | *Nomia melanderi* |
| | | 1.5 | >8 | 1 | | *Apis mellifera* |
| | | | >8 | 1 | | *Megachile rotundata* |
| | | | >8 | 1 | | *Nomia melanderi* |
| | | 3 | >8 | 1 | | *Apis mellifera* |
| | | | >8 | 1 | | *Megachile rotundata* |
| | | | >8 | 1 | | *Nomia melanderi* |
| | WP | 0.125 | >18 | 1 | <42 | *Apis mellifera* |
| | | 0.25 | >18 | 1 | <42 | |
| | | 0.5 | >42 | 1 | >42 | |
| | | 1 | >42 | 1 | >42 | |
| Dicofol | EC | 1.5 | <3 | 1 | | *Apis mellifera* |
| Dimethoate | EC | 0.125 | <3 | 1 | | *Apis mellifera* |
| | | | <3 | 1 | | *Megachile rotundata* |
| | | 0.25 | 4.18 | 1 | | *Apis mellifera* |
| | | | 3 | 1 | | *Megachile rotundata* |
| | | 0.5 | 114 or 11.9 | 2 | <120 | *Apis mellifera* |
| | | | 121 | 2 | <120 | *Megachile rotundata* |
| | | | >72 | 2 | >72 | *Nomia melanderi* |
| Disulfoton | EC | 0.5 | <3 | 1 | | *Apis mellifera* |
| | | | 13 | 1 | | *Megachile rotundata* |
| | | | <3 | 1 | | *Nomia melanderi* |
| | | 1 | 8.86 | 1 | 5.5 | *Apis mellifera* |
| | | | 20.7 | 1 | | *Megachile rotundata* |
| | | | 2.23 | 1 | | *Nomia melanderi* |
| Endosulfan | EC | 0.75 | <2 | 1 | <3 | *Apis mellifera* |
| | | | >8 | 1 | | *Megachile rotundata* |
| | | | 6.75 | 1 | | *Nomia melanderi* |
| | WP | 0.5 | <8 | 1 | | *Apis mellifera* |
| | | | >8 | 1 | | *Megachile rotundata* |
| | | 0.75 | >8 | 1 | | *Megachile rotundata* |
| | | 1 | >8 | 1 | | |

**Table 2** (*continued*)

| Active Ingredient[1] | Formulation | Rate (lb ai/A) | Calculated RT$_{25}$ Values | | | Test Species Scientific Name |
|---|---|---|---|---|---|---|
| | | | **Meta-Analysis** | **# of Studies** | **EPA Reported Value** | |
| Esfenvalerate | EC | 0.0125 | <2 | 1 | | *Apis mellifera* |
| | | | >8 | 2 | | *Apis mellifera* |
| | | 0.05 | <2 | 2 | | *Megachile rotundata* |
| | | | 8 | 1 | | *Nomia melanderi* |
| | | 0.075 | >8 | 1 | | *Apis mellifera* |
| | | | <2 | 1 | | *Megachile rotundata* |
| | | 0.1 | >24 | 2 | | *Apis mellifera* |
| | | | <2 | 1 | | *Megachile rotundata* |
| Ethiprole | EC | 0.18 | 643 | 1 | | *Apis mellifera* |
| | SC | 0.3 | 333 | 1 | | |
| Fenitrothion | EC | 0.5 | 18.2 | 2 | <24 | *Apis mellifera* |
| | | | >72 | 1 | 106 | *Megachile rotundata* |
| | | | >72 | 1 | 98 | *Nomia melanderi* |
| | | 1 | >72 | 2 | 101 | *Apis mellifera* |
| | | | >120 | 1 | >120 | *Megachile rotundata* |
| | | | >120 | 1 | >120 | *Nomia melanderi* |
| Fenpropathrin | EC | 0.1 | >8 | 1 | <192 | *Apis mellifera* |
| | | | >8 | 1 | | *Megachile rotundata* |
| | | | >8 | 1 | | *Nomia melanderi* |
| | | 0.2 | >8 | 2 | 276 | *Apis mellifera* |
| | | | >8 | 1 | | *Megachile rotundata* |
| | | | >8 | 1 | | *Nomia melanderi* |
| | | 0.4 | >8 | 2 | <336 | *Apis mellifera* |
| | | | >8 | 1 | | *Megachile rotundata* |
| | | | >8 | 1 | | *Nomia melanderi* |
| Fenvalerate | EC | 0.1 | 6.5 | 2 | 7 | *Apis mellifera* |
| | | | >8 | 1 | >8 | *Megachile rotundata* |
| | | | 6.82 | 2 | 7 | *Nomia melanderi* |
| | | 0.2 | 16.4 | 1 | | *Apis mellifera* |
| | | | >8 | 1 | | *Megachile rotundata* |
| | | 0.4 | >8 | 1 | >8 | *Apis mellifera* |
| | | | >8 | 1 | >8 | *Megachile rotundata* |
| | | | >8 | 1 | >8 | *Nomia melanderi* |

**Table 2** (*continued*)

| Active Ingredient[1] | Formulation | Rate (lb ai/A) | Calculated RT$_{25}$ Values | | | Test Species Scientific Name |
|---|---|---|---|---|---|---|
| | | | Meta-Analysis | # of Studies | EPA Reported Value | |
| | SC | 0.01 | 238 | 1 | | *Apis mellifera* |
| | | | <2 | 1 | | *Apis mellifera* |
| | | 0.0125 | 3.82 | 1 | | *Megachile rotundata* |
| | | | <2 | 1 | | *Nomia melanderi* |
| | | | <2 | 1 | | *Apis mellifera* |
| | | 0.025 | <2 | 1 | | *Megachile rotundata* |
| | | | <2 | 1 | | *Nomia melanderi* |
| | | | 7.15 | 1 | | *Apis mellifera* |
| | | 0.1 | >8 | 1 | | *Megachile rotundata* |
| | | | <2 | 1 | | *Nomia melanderi* |
| | | | >8 | 1 | | *Apis mellifera* |
| Fipronil | | 0.2 | >8 | 1 | | *Megachile rotundata* |
| | | | <2 | 1 | | *Nomia melanderi* |
| | | | <2 | 1 | | *Apis mellifera* |
| | | 0.0125 | <2 | 1 | | *Megachile rotundata* |
| | | | <2 | 1 | | *Nomia melanderi* |
| | | | <2 | 1 | | *Apis mellifera* |
| | | 0.025 | <2 | 1 | | *Megachile rotundata* |
| | | | <2 | 1 | | *Nomia melanderi* |
| | WG | | 5.51 or >8 | 2 | | *Apis mellifera* |
| | | 0.1 | >8 or 3.52 | 2 | | *Megachile rotundata* |
| | | | <2 | 2 | | *Nomia melanderi* |
| | | | >8 | 2 | | *Apis mellifera* |
| | | 0.2 | >8 | 1 | | *Megachile rotundata* |
| | | | <2 | 1 | | *Nomia melanderi* |
| Fluazinam | WDG | 0.135 | <2 | 1 | | *Megachile rotundata* |
| Flupyradifurone | SL | 0.183 | <3 | 1 | <3 | *Apis mellifera* |
| Fluvalinate | E | 0.1 | <2 | 1 | | *Apis mellifera* |
| Fonofos | Enc. | 1 | <3 | 1 | | *Apis mellifera* |
| | | 2 | >8 | 1 | | |
| | EC | 1 | <3 | 1 | <3 | |
| | | 2 | 5.76 | 1 | <8 | |
| | | | <3 | 1 | | *Apis mellifera* |
| | | 0.23 | <3 | 1 | | *Megachile rotundata* |
| | | | <3 | 1 | | *Nomia melanderi* |
| | | | <3 | 1 | | *Apis mellifera* |

**Table 2** (*continued*)

| Active Ingredient[1] | Formulation | Rate (lb ai/A) | Calculated RT$_{25}$ Values | | | Test Species Scientific Name |
|---|---|---|---|---|---|---|
| | | | Meta-Analysis | # of Studies | EPA Reported Value | |
| Formetanate Hydrochloride | SP | 0.45 | <3 | 1 | | *Megachile rotundata* |
| | | | <3 | 1 | | *Nomia melanderi* |
| | | | <2 | 4 | | *Apis mellifera* |
| | | 0.5 | <3, 7.5, or >8 | 4 | | *Megachile rotundata* |
| | | | 11.2 or <3 | 3 | | *Nomia melanderi* |
| | | | 4.32 | 3 | | *Apis mellifera* |
| | | 1 | 5.3 | 2 | | *Megachile rotundata* |
| | | | 5.15 | 1 | | *Nomia melanderi* |
| | | | 6.68 | 1 | | *Apis mellifera* |
| | | 1.1 | >8 | 1 | | *Megachile rotundata* |
| | | | >8 | 1 | | *Nomia melanderi* |
| Imidacloprid | EC | | 90 | 1 | | *Apis mellifera* |
| | | 0.25 | 214 | 1 | | *Megachile rotundata* |
| | | | >72 | 1 | | *Nomia melanderi* |
| | | | 110 | 1 | | *Apis mellifera* |
| | | 0.5 | 277 | 2 | | *Megachile rotundata* |
| | | | >72 | 1 | | *Nomia melanderi* |
| | F | | <2 | 1 | | *Apis mellifera* |
| | | 0.15 | >8 | 1 | | *Megachile rotundata* |
| | | | 2.72 | 1 | | *Nomia melanderi* |
| | | 0.1 | 2.56 | 1 | <8 | *Nomia melanderi* |
| | SL | 0.018 | 236 | 1 | | *Apis mellifera* |
| | | 0.045 | <3 | 1 | | *Apis mellifera* |
| | WG | 0.167 | 31.1 | 1 | | |
| | | 0.5 | 89.8 | 1 | | |
| Indoxacarb | SC | 0.039 | 140 | 1 | | *Apis mellifera* |
| Lambda-cyhalothrin | E | 0.02 | 17 | 1 | | *Apis mellifera* |
| | | 0.03 | >24 | 1 | | *Apis mellifera* |
| | | | >8 | 1 | | *Megachile rotundata* |
| | EC | 0.01 | 54 | 1 | | *Apis mellifera* |
| | | | >72 | 1 | | *Megachile rotundata* |
| | | 0.02 | >72 | 1 | | *Apis mellifera* |
| | | | >72 | 1 | | *Megachile rotundata* |
| Leptophos | EC | 1 | 2.32 | 2 | | *Apis mellifera* |
| | | | 13.8 | 2 | | *Megachile rotundata* |
| | | | 3.86 | 2 | | *Nomia melanderi* |
| | | 2 | >8 | 1 | | *Apis mellifera* |
| | | 0.5 | >8 | 1 | 24 | |

**Table 2** (*continued*)

| Active Ingredient[1] | Formulation | Rate (lb ai/A) | Calculated RT$_{25}$ Values | | | Test Species Scientific Name |
|---|---|---|---|---|---|---|
| | | | Meta-Analysis | # of Studies | EPA Reported Value | |
| Lindane | EC | 1 | >24 | 1 | 72 | *Apis mellifera* |
| | | 1.5 | >48 | 1 | 72 | |
| | F | 0.5 | >8 | 1 | 24 | |
| | | 1 | >48 | 1 | 72 | |
| | | 1.5 | >72 | 1 | 72 | |
| | WP | 0.5 | >8 | 1 | 24 | |
| | | 1 | >48 | 1 | 72 | |
| | | 1.5 | >48 | 1 | 72 | |
| Malathion | E | 1 | >8 | 1 | | *Apis mellifera* |
| | | | >8 | 1 | | *Megachile rotundata* |
| | | | >8 | 1 | | *Nomia melanderi* |
| | EC | 0.625 | >18 | 1 | | *Apis mellifera* |
| | | 1 | >24 | 1 | | |
| | | 1.25 | >42 | 1 | | |
| | WP | 0.3125 | >18 | 1 | | *Apis mellifera* |
| | | 0.625 | >42 | 1 | | |
| | | 1.25 | >42 | 1 | | |
| Malonoben | EC | 0.5 | <8 | 1 | | *Megachile rotundata* |
| | | | <2 | 1 | | *Nomia melanderi* |
| | | 1 | >8 | 1 | | *Megachile rotundata* |
| | | | <2 | 1 | | *Nomia melanderi* |
| | | 2 | >24 | 1 | | *Megachile rotundata* |
| | | | >8 | 1 | | *Nomia melanderi* |
| | WP | 0.25 | <2 | 1 | | *Apis mellifera* |
| | | | <2 | 1 | | *Megachile rotundata* |
| | | | <2 | 1 | | *Nomia melanderi* |
| | | 0.5 | <2 | 1 | | *Apis mellifera* |
| | | | 6 | 1 | | *Megachile rotundata* |
| | | | <2 | 1 | | *Nomia melanderi* |
| | | 1 | <2 | 1 | | *Apis mellifera* |
| | | | 18 | 1 | | *Megachile rotundata* |
| | | | <2 | 1 | | *Nomia melanderi* |
| Methamidophos | EC | 0.67 | >8 | 1 | | *Apis mellifera* |
| | | | >8 | 1 | | *Megachile rotundata* |
| | | | >8 | 1 | | *Nomia melanderi* |
| Methidathion | E | 0.736 | >8 | 1 | | *Apis mellifera* |
| | | | >8 | 1 | | *Megachile rotundata* |
| | | | >8 | 1 | | *Nomia melanderi* |
| | EC | 1 | 91 | 1 | | *Apis mellifera* |
| | | | 89.6 | 1 | | *Megachile rotundata* |
| | | | >72 | 1 | | *Nomia melanderi* |

**Table 2** (*continued*)

| Active Ingredient[1] | Formulation | Rate (lb ai/A) | Calculated RT$_{25}$ Values | | | Test Species Scientific Name |
|---|---|---|---|---|---|---|
| | | | **Meta-Analysis** | **# of Studies** | **EPA Reported Value** | |
| Methomyl | EC | 0.9 | <2 | 1 | | *Apis mellifera* |
| | | | <3 | 1 | | *Apis mellifera* |
| | | 0.25 | <4 | 1 | | *Megachile rotundata* |
| | | | <4 | 1 | | *Nomia melanderi* |
| | | | <3 | 1 | | *Apis mellifera* |
| | LS | 0.5 | <4 | 1 | | *Megachile rotundata* |
| | | | 5 | 1 | | *Nomia melanderi* |
| | | | 6.11 | 1 | | *Apis mellifera* |
| | | 1 | 20.5 | 1 | | *Megachile rotundata* |
| | | | >24 | 1 | | *Nomia melanderi* |
| | | | <2 | 1 | | *Apis mellifera* |
| | | 0.5 | 5.2 | 1 | | *Megachile rotundata* |
| | | | 4.53 | 1 | | *Nomia melanderi* |
| | | | >8 | 1 | | *Apis mellifera* |
| | WP | 0.9 | >8 | 1 | | *Megachile rotundata* |
| | | | >8 | 1 | | *Nomia melanderi* |
| | | | <8 | 1 | | *Apis mellifera* |
| | | 1 | 5.87 | 1 | | *Megachile rotundata* |
| | | | 6 | 1 | | *Nomia melanderi* |
| Methyl Parathion | CS | 0.401 | 205 | 1 | 207 | *Apis mellifera* |
| | | | 76 | 3 | | *Apis mellifera* |
| | | 0.5 | >72 | 2 | | *Megachile rotundata* |
| | EC | | >8 | 1 | | *Nomia melanderi* |
| | | 1 | 81 | 2 | | *Apis mellifera* |
| | | | >72 | 1 | | *Megachile rotundata* |
| | | | >8 | 1 | | *Apis mellifera* |
| | F | 0.5 | >8 | 1 | | *Megachile rotundata* |
| | | | >8 | 1 | | *Nomia melanderi* |
| Naled | | | >8 or <8 | 2 | | *Apis mellifera* |
| | E | 1 | >8 | 1 | | *Megachile rotundata* |
| | | | 2 | 1 | | *Nomia melanderi* |
| | | | >8 | 2 | | *Apis mellifera* |
| | EC | 1 | 6.44 or >72 | 2 | | *Megachile rotundata* |
| | | | >24 | 1 | | *Nomia melanderi* |

**Table 2** (*continued*)

| Active Ingredient[1] | Formulation | Rate (lb ai/A) | Calculated RT$_{25}$ Values | | | Test Species Scientific Name |
|---|---|---|---|---|---|---|
| | | | Meta-Analysis | # of Studies | EPA Reported Value | |
| | EC | 1 | >24 | 1 | | *Apis mellifera* |
| | | | <4 | 1 | | *Apis mellifera* |
| | | 0.25 | <4 | 1 | | *Megachile rotundata* |
| | | | <4 | 1 | | *Nomia melanderi* |
| Oxamyl | | | <4 | 1 | | *Apis mellifera* |
| | LS | 0.5 | >9 | 1 | | *Megachile rotundata* |
| | | | >9 | 1 | | *Nomia melanderi* |
| | | | 12.5 | 1 | 22 | *Apis mellifera* |
| | | 1 | >24 | 1 | | *Megachile rotundata* |
| | | | >24 | 1 | | *Nomia melanderi* |
| | | 0.5 | <2 | 1 | | *Apis mellifera* |
| | | | <2 | 1 | | *Megachile rotundata* |
| | EC | 0.75 | <2 | 1 | | *Apis mellifera* |
| Oxydemeton-methyl | | | <2 | 1 | | *Megachile rotundata* |
| | | 1 | 6 | 1 | | *Nomial melanderi* |
| | | | <2 | 1 | | *Apis mellifera* |
| | SC | 0.5 | <2 | 1 | | *Megachile rotundata* |
| | | | <2 | 1 | | *Nomia melanderi* |
| | | | 12.6 | 1 | | *Apis mellifera* |
| Parathion | EC | 0.5 | 11.5 | 1 | | *Megachile rotundata* |
| | | | 12.8 | 1 | | *Nomia melanderi* |
| | | | 21 | 1 | | *Apis mellifera* |
| | | 0.05 | >24 | 1 | | *Megachile rotundata* |
| | | | 15 | 1 | | *Nomia melanderi* |
| | | | 169 | 3 | | *Apis mellifera* |
| | EC | 0.1 | >24 | 1 | | *Megachile rotundata* |
| | | | >24 | 1 | | *Nomia melanderi* |
| | | 0.125 | >8 | 1 | | *Megachile rotundata* |
| Permethrin | | | >168 | 2 | | *Apis mellifera* |
| | | 0.2 | >24 | 1 | | *Megachile rotundata* |
| | | | >24 | 1 | | *Nomia melanderi* |
| | ULV | 0.1 | 95.3 | 1 | | *Apis mellifera* |
| | | 0.05 | >72 | 1 | | *Apis mellifera* |
| | WP | | >72 | 1 | | *Megachile rotundata* |
| | | 0.1 | >72 | 1 | | *Apis mellifera* |
| | | | >72 | 1 | | *Megachile rotundata* |
| | | 0.15625 | 18 | 1 | | |
| Phenthoate | EC | 0.3125 | >18 | 1 | | *Apis mellifera* |
| | | 0.625 | >42 | 1 | | |
| | | 1.25 | >42 | 1 | | |

**Table 2** (*continued*)

| Active Ingredient[1] | Formulation | Rate (lb ai/A) | Calculated RT$_{25}$ Values | | | Test Species Scientific Name |
|---|---|---|---|---|---|---|
| | | | **Meta-Analysis** | **# of Studies** | **EPA Reported Value** | |
| Phosmet | EC | 1 | >8 | 1 | | *Apis mellifera* |
| | | 2 | >8 | 1 | | |
| | WP | 1 | >8 | 1 | >3 | |
| | | 2 | >8 | 1 | | |
| Prochloraz | EC | 0.5 | <2 | 1 | | *Apis mellifera* |
| | | 1 | <2 | 1 | | |
| | | 2 | <2 | 1 | | |
| Profenofos | EC | 1 | >8 | 1 | | *Apis mellifera* |
| | | | >8 | 1 | | *Megachile rotundata* |
| | | | >8 | 1 | | *Nomia melanderi* |
| Propargite | EC | 2.1 | <3 | 1 | | *Apis mellifera* |
| | | | <3 | 1 | | *Megachile rotundata* |
| | | 2.25 | <3 | 1 | | *Apis mellifera* |
| | | | <3 | 1 | | *Megachile rotundata* |
| Piperonyl butoxide | E | 0.5 | >24 | 1 | | *Apis mellifera* |
| Pyrethrins | EC | 1 | <2 | 1 | | *Apis mellifera* |
| | | | <2 | 1 | | *Megachile rotundata* |
| | | | <2 | 1 | | *Nomia melanderi* |
| Sulfloxaflor | SC | 0.18 | <1 | 3 | | *Megachile rotundata* |
| | | | <1 | 3 | | *Nomia melanderi* |
| Tetraniliprole | SC | 0.027 | <3 | 1 | | *Apis mellifera* |
| | | 0.054 | <3 | 1 | | |
| | | 0.089 | <3 | 1 | | |
| Thiacloprid | SC | 0.045 | <2 | 1 | | *Apis mellifera* |
| | | 0.09 | <2 | 1 | | |
| | | 0.16 | <2 | 1 | <2 | |
| Thiodicarb | F | 0.5 | <2 | 1 | | *Apis mellifera* |
| | | 1.2 | 77 | 1 | | |
| | WDG | 1 | >8 | 1 | | *Apis mellifera* |
| Tiazamate | E | 0.25 | <2 | 1 | | *Apis mellifera* |
| | | | <2 | 1 | | *Nomia melanderi* |
| Tolfenpyrad | EC | 1.69 | >168 | 1 | | *Megachile rotundata* |
| | | | >168 | 1 | | *Nomia melanderi* |
| Trichlorfon | SP | 1 | <8, or >8 or 5.39 | 5 | | *Apis mellifera* |
| | | | 4.45 | 3 | | *Megachile rotundata* |
| | | | 4.64 | 2 | | *Nomia melanderi* |

**Table 2** (*continued*)

| Active Ingredient[1] | Formulation | Rate (lb ai/A) | Calculated RT$_{25}$ Values | | | Test Species Scientific Name |
|---|---|---|---|---|---|---|
| | | | Meta-Analysis | # of Studies | EPA Reported Value | |
| | EW | 0.037 | >8 | 1 | | *Apis mellifera* |
| | | | >8 | 1 | | *Megachile rotundata* |
| Zeta-cypermethrin | | | >8 | 1 | | *Nomia melanderi* |
| | | | >72 | 1 | | *Apis mellifera* |
| | WP | 1 | >8 | 1 | | *Megachile rotundata* |
| | | | >8 | 1 | | *Nomia melanderi* |

**Notes.**

E, emulsifiable; EC, emulsifiable concentrate; Enc., encapsulated; EW, emulsion in water; F, flowable; LS, liquid soluble; SC, soluble concentrate; SL, soluble (liquid) concentrate; SP, soluble powder; ULV, ultra-low volume liquid; WDG, water dispersible granular; WG, wettable granule; WP, wettable powder.

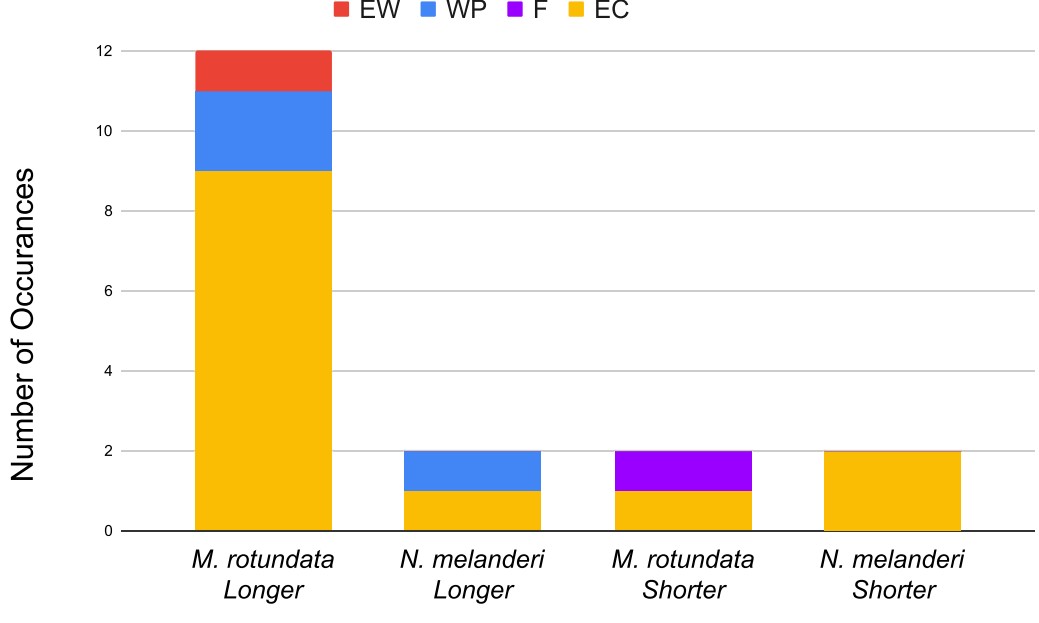

**Figure 3   Comparison of bee species across different active ingredient formulations.** EC (emulsifiable concentrate), F (flowable), WP (wettable powder), EW (emulsion in water), LS (liquid soluble), and SC (soluble concentrate). *M. rotundata* and *N. melanderi* RT25 values were compared to *A. mellifera* and reported as either (1) longer than *A. mellifera* values ("*M. rotundata* longer" and "*N. melanderi* longer") or (2) shorter than *A. mellifera* values ("*M. rotundata* Shorter" and "*N. melanderi* Shorter").

of extended residual toxicity, even though the actual estimate of RT$_{25}$ beyond the 8 h threshold remains unresolved. Nevertheless, the general agreement between studies on extended residual toxicity is remarkable and suggests that RT$_{25}$ estimates are relatively insensitive to variation in lab technique and weathering conditions.

Our preliminary finding that lab methodology and field weathering conditions are not important sources of variation for RT$_{25}$ should be confirmed experimentally. With respect to lab methodology, we think three factors warrant closer examination, namely the

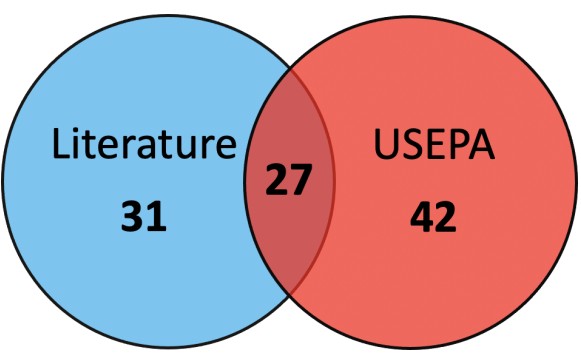

**Figure 4** **The source of calculated RT$_{25}$ values for pesticide active ingredients.** Number of pesticide active ingredients where RT$_{25}$ values could only be calculated from the literature ("Literature"), only from the EPA's published database ("USEPA"; *United States Environmental Protection Agency, 2014*) or there were RT$_{25}$ values available from both ("Both").

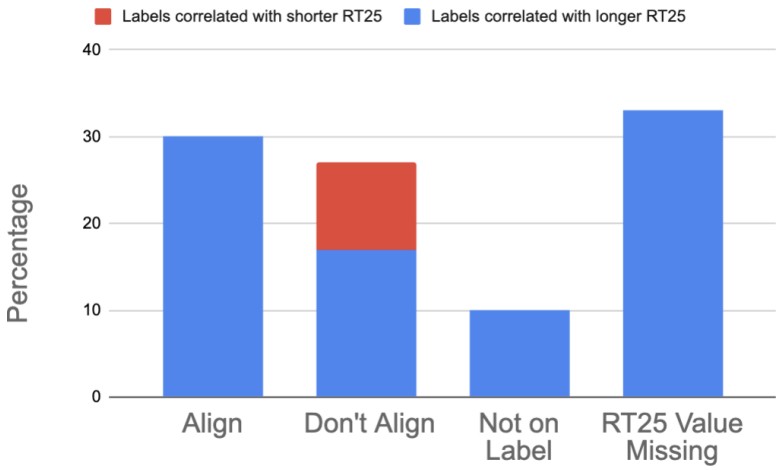

**Figure 5** **Comparison of pesticide label language indicating residual toxicity in relation to RT$_{25}$ values (calculated from the literature and from the EPA** *United States Environmental Protection Agency, 2014*). Residual toxicity language in the Environmental Hazards section either: (1) aligned with RT$_{25}$ values ("Align"), (2) did not align ("Don't Align"), (3) lacked residual toxicity language ("Not on Label") or (4) did not have an RT$_{25}$ value to relate to the label language ("RT$_{25}$ value missing"). Formulation was matched when comparing label language to calculated RT$_{25}$ values.

temperature at which the assay is performed, the number of bees held in each test cage and the age of bee used in the test. We report considerable variation in the temperature bees are exposed to in test cage, with temperatures tending to be lower on average compared to EPA guidance. Cooler temperatures could decrease bee activity, leading to less overall contact with the pesticide residue and shorter residual toxicity values (*Corbet et al., 1993*). The number of bees in test cages may also influence RT$_{25}$ values by concentrating/diluting the pesticide across fewer/greater numbers of bees, resulting in shorter/longer RT$_{25}$ values. For example, a cage with 500 bees walking over 10g of pesticide contaminated leaf material may ultimately receive a lower dose per bee than if only 10 bees were walking over the same

material over the same period of time. We observed that *M. rotundata* and *N. melanderi* had, on average, fewer bees per cage compared to *A. mellifera* which could lead to more contact per bee to the pesticide residues. Most studies deviated from the age of bees recommended by the EPA; however, using less than one day old bees may be distorting, as foraging age bees, which are typically bees that are least three weeks old, are the bees likely to contact weathered residues in the field. Notably, a factor that was largely omitted from most studies was a description of the weathering conditions, such as temperature, humidity, precipitation, and cloud cover. Potentially, weathering conditions may have a larger impact on $RT_{25}$ estimates than variation in laboratory methodology.

We observed trends in $RT_{25}$ values among different rates and formulations of active ingredients. Typically, the higher the application rate of a pesticides, the longer the calculated $RT_{25}$ values. For example, the calculated $RT_{25}$ value for the organophosphate insecticide chlorpyrifos emulsifiable concentrate with *A. mellifera* was 17 h at the rate of 0.25 lb ai/A and 99 h $RT_{25}$ time at 0.5 lb ai/A. This suggests that $RT_{25}$ may be different for different application rates, which draws into question the premise of the Pollinating Insect Hazard Statement, where a single residual toxicity statement is meant to cover multiple different use patterns of a pesticide, such as different rates. Notably, new guidance issued by EPA (2017) moves away from relying on the Pollinating Insect Hazard Statement to convey residual toxicity estimates, relying more on specific use directions, where rate and crop are specified. Our results suggest this shift will provide applicators with more guidance on the specific residual times they might experience in the field.

The species of bee used to estimate $RT_{25}$ exhibited notable patterns that should be further investigated. In general, we observed that for the same active ingredient applied at the same rate and formulation *M. rotundata* had longer $RT_{25}$ times compared to *A. mellifera*, and that *N. melanderi* had both shorter and longer $RT_{25}$ times compared to *A. mellifera*. Emulsifiable concentrates were associated with the largest difference in $RT_{25}$ estimates among species, with *M. rotundata* consistently having longer $RT_{25}$ values than *A. mellifera* for these formulations. It is unclear what is the source of these patterns. One hypothesis is that *M. rotundata* may be more susceptible to pesticides as this species lacks the ability to detoxify certain synthetic insecticides that are normally metabolized by other bee species (*Hayward et al., 2019*). Certainly, several studies have indicated differential toxicity of pesticides to different bee taxa (*Johansen et al., 1983a*; *Johansen et al., 1983b*; *Mayer, Kovacs & Lunden, 1998*; *Devillers & Pham-Delegue, 2002*). Another possible explanation for the difference between bee species could be their size difference. *M. rotundata* has the smallest average size of the three bees we analyzed and, therefore, would have the highest ratio of surface area to body volume. The higher surface area to body volume ratio results in a higher rate in the accumulation of lethal dosages over time (*Johansen et al., 1983a*; *Johansen et al., 1983b*; *Wisk et al., 2014*) potentially resulting in longer $RT_{25}$ times for smaller bees, for a given toxicity of a pesticide. Little research has been done into the effects of differing formulations on the residual toxicity across bee species. A species comparative study would be useful to determine what variables (*e.g.*, differences in behavior, different physiology, etc.) contribute to the differing residual toxicity values. Currently, the EPA publicly reports (*United States Environmental Protection Agency, 2014*) $RT_{25}$ times primarily for
*A. mellifera*, with limited data available on other species of pollinating bees. Researchers also primarily use *A. mellifera* when conducting pesticide risk assessments which lead to large knowledge gaps for other pollinating bees impacted by pesticides (*Tosi et al., 2022*). Differences in species residual toxicity times have been noticed in the past (*Johansen et al., 1983a*; *Johansen et al., 1983b*; *Mayer, Lunden & Jasso, 1997*) and variation in pesticide sensitivity among bee species has been shown which could suggest variation in residual toxicity times (*Arena & Sgolastra, 2014*). However, there have been no in-depth studies designed to comparatively characterize $RT_{25}$ estimates for different species, let alone resolve the mechanisms by which bees may respond to the dissipation of residues differently. Our results suggest that honey bee residual toxicity assay results may not be generalizable to other bee species as has been done in the past. Variation in $RT_{25}$ estimates for different bee species would be important information for pesticide applicators, particularly if they are using residual times for bee species with the shortest $RT_{25}$ values.

The finding from our study that is of greatest concern to pesticide applicators was widespread misalignment between $RT_{25}$ values and statements of residual toxicity in the Pollinating Insect Hazard Statement. Of the pesticide labels we were able to compare to calculated $RT_{25}$ values, almost a third were inaccurate in the wording of their Pollinating Insect Hazard Statement. For example, the formulated end-use product Perm-Up 3.2 EC (USEPA registration number 70506-9) containing the pyrethroid insecticide permethrin indicates the product should not be applied while bees are "actively visiting" suggesting a less than 8-hour residual toxicity time. However, the residual toxicity studies for permethrin consistently indicated $RT_{25}$ values greater than 8 h even at the lowest application rate calculated, 0.05 lb ai/A. Although this finding is concerning, some of these discrepancies may arise from our assumption that all pesticides with the same active ingredient and applied at the same rate have similar $RT_{25}$ values. Potentially, pesticide products may have different residual times owing to features independent of the active ingredient, such as inert ingredients. Our assumption that $RT_{25}$ can be generalized across products containing the same active ingredient is supported by our findings that $RT_{25}$ estimates were largely consistent for active ingredients across studies and relative to estimates published by EPA (*United States Environmental Protection Agency, 2014*). Nevertheless, we suggest caution in interpreting our results since the number of different products used to estimate $RT_{25}$ values for each active ingredient tended to be dwarfed by the total number of registered products containing those ingredients on the market. Regardless, our study indicates that either there is high variability in residual toxicity between pesticides containing the same active ingredients, which calls into the question efforts like the EPA's to publish $RT_{25}$ values based on active ingredients, or the Pollinating Insect Hazard Statement on existing pesticide labels aligns poorly with $RT_{25}$ values. Our data currently suggests the latter problem predominates, resulting in pesticide applicators lacking a reliable piece of information to mitigate exposure to bees during bloom.

One thing is clear from our study: there remain large gaps in our database of $RT_{25}$ estimates. Although this database is the most comprehensive to date, and expands on published values by the EPA, the lack of publicly accessible $RT_{25}$ estimates is something we hope researchers will make a concerted effort to address. We also encourage the EPA

to review its existing data from registrants, which is unavailable to researchers, pesticide applicators and the public, and fill gaps in its public-facing database. Alternatively, the EPA could develop a mechanism to release registrant-collected residual toxicity data to the public to enable researchers to develop such a database independently. While estimating residual toxicity has been a part of the pesticide risk assessment process for decades, its relevance continues with new guidance around label language that foregrounds $RT_{25}$ values beyond the Environmental Hazard section to the crop-specific directions for use on the label (*United States Protection Agency, 2017*). The need to create a basis for evaluation of these changes is not only important for pesticide applicators who are seeking instruction to protect bees from exposure, but for the sustainable management of domesticated managed bee stocks and wild bee communities.

## CONCLUSIONS

Through our efforts, we were successfully able to create a compendium of $RT_{25}$ values that could be used to determine if pesticide label language aligns with calculated active ingredient $RT_{25}$ values. There was noticeable variation in species and application rate which could call into question whether a single Pollinating Insects Hazards Statement is sufficient to fully communicate the hazards of a pesticide product. Further comparison of the calculated values to published EPA values revealed that lab methodology does not seem to affect $RT_{25}$ values as seen from comparison of study values to the EPA, though field conditions during the weathering of the pesticide may need to be explored further. Comparing a combined database of published EPA values and our calculated $RT_{25}$ values to label language showed significant misalignment in Pollinating Insect Hazard Statements. The variation in residual toxicity remains an emerging field of research that must be addressed to ensure the applications of pesticides is occurring in a safe manner to minimize the risk towards pollinating bees.

## ACKNOWLEDGEMENTS

We thank Drs. Daniel Schmehl and Allen Olmstead at Bayer CropScience for unpublished residual toxicity data and technical assistance with calculations of $RT_{25}$ as well as Dr. Theresa Pitts-Singer for compiling residual toxicity studies from the Western Alfalfa Seed Growers Association. We also owe a debt of gratitude to Dr. Thomas Steeger for reviewing an advanced draft of this manuscript.

### Funding

The research was supported from grants from the Western IPM Center (No. 2018-70006-28881) and Western Sustainable Agriculture and Education Research and Education (No. G258-19-W7500). The funders had no role in study design, data collection and analysis, decision to publish, or preparation of the manuscript.

## Grant Disclosures

The following grant information was disclosed by the authors:

Western IPM Center: 2018-70006-28881.

Western Sustainable Agriculture and Education Research and Education: G258-19-W7500.

## Competing Interests

The authors declare there are no competing interests.

## Author Contributions

- Leah Swanson conceived and designed the experiments, performed the experiments, analyzed the data, prepared figures and/or tables, authored or reviewed drafts of the article, and approved the final draft.
- Andony Melathopoulos conceived and designed the experiments, authored or reviewed drafts of the article, and approved the final draft.
- Matthew Bucy analyzed the data, authored or reviewed drafts of the article, and approved the final draft.

## Data Availability

The raw data is available in the Supplementary File.

## Supplemental Information

Supplemental information for this article can be found online at http://dx.doi.org/10.7717/peerj.16672#supplemental-information.

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

## FURTHER READING

Akca I, Tuncer C, Guler A, Sarugan I. 2009. Residual toxicity of 8 different insecticides on honey bee (*Apis mellifera* Hymenoptera: Apidae). *Journal of Animal and Veterinary Advances* **8**:436–440.

Bailey J, Scott-Dupree C, Harris R, Tolman J, Harris B. 2005. Contact and oral toxicity to honey bees (*Apis mellifera*) of agents registered for use for sweet corn insect control in Ontario, Canada. *EDP Sciences* **36**:623–633 DOI 10.1051/apido:2005048.

Clinch P. 1967. The residual contact toxicity to honey bees of insecticides sprayed on to white clover (*Trifolium repens* I.) in the laboratory. *New Zealand Journal of Agricultural Research* **10**:289–300 DOI 10.1080/00288233.1967.10425136.

Johansen C. 1972. Toxicity of field-weathered insecticide residues to four kinds of bees. *Environmental Entomology* **1**:393–394 DOI 10.1093/ee/1.3.393.

Johansen C, Baird C. 1972. Small-scale bee poisoning tests with honey bees (HB), alkali bees (AB), and alfalfa leafcutting bees (LB). In: *Bee research investigations*. 2–3. Pullman: Washington State University, 13–17.

Johansen C, Eves J. 1971. Small-scale bee poisoning tests with honey bees (HB), alkali bees (AB), alfalfa leafcutter bees (LB), and bumble bees (BB). In: *Bee research investigations*. 2–3. Pullman: Washington State University, 13–17.

Johansen C, Kious C, Schultz George, Gupta R, Madsen R, Robinson W. 1977. Investigation of the bee poisoning hazards of microencapsulated methyl parathion (penncap-M). In: *Bee research investigations*. 1-5. Pullman: Washington State University, 16, 19–20.

Johansen C, Mayer D, Baird C. 1973. Small-scale bee poisoning tests with honey bees (HB), alkali bees (AB), and alfalfa leafcutting bees (LB). In: *Bee research investigations*. 3-4. Pullman: Washington State University, 9–14.

Johansen C, Mayer D, Kious C. 1984. Small-scale poisoning tests with honey bees and alfalfa leafcutting bees. In: *Bee research investigations*. Pullman: Washington State University, 1–2.

Johansen C, Mayer D, Madsen R, Robinson W. 1975. Small-scale bee poisoning tests with honey bees (HB), alkali bees (AB), and alfalfa leafcutting bees (LB). In: *Bee research investigations*. 1–2. Pullman: Washington State University, 12–15.

Johansen C, Mayer D, Robinson W, Gupta R, Spann J, Madsen R. 1976. Small-scale bee poisoning tests with honey bees (HB), alkali bees (AB), and alfalfa leafcutting bees (LB). In: *Bee research investigations*. 1-2. Pullman: Washington State University, 13–16.

**Keshlaf M, Basta A, Spooner-Hart R. 2013.** Assessment of toxicity of fipronil and its residues to honey bees. *Mellifera* **13**:30–38.

**Kim B-S, Park Y-K, Lee Y-H, Joeng M-H, You A-S, Yang Y-J, Kim J-B, Kwon O-K, Ahn Y-J. 2008.** Honeybee acute and residual toxicity of pesticides registered for strawberry. *The Korean Journal of Pesticide Science* **12**:229–235.

**Kious C, Schultz G, Johansen C. 1979.** Small-scale bee poisoning tests with honey bees (*Apis mellifera*). In: *Bee research investigations.* Pullman: Washington State University, 3–4.

**Mayer DF, Johansen C. 1985.** Pollinator protection and Acephate (Orthene) insecticide. *Agricultural Research* **125**:207–210.

**Mayer D, Johansen C, Shanks C, Lunden J.** Insecticide residues. In: *Methomyl and honey bees.* Prosser: Washington State University, 46–47.

**Mayer D, Johansen C, Shanks C, Pike K. 1987a.** Effects of Fenvalerate Insecticide on Pollinators. *Journal of Entomological Society of British Columbia* **84**:39–45.

**Mayer D, Kovacs G, Brett B, Bisabri B. 2001.** The effects of spinosad insecticide to adults of *Apis mellifera*, Megachile rotundata and *Nomia melanderi* (Hymenoptera: Apidae). *International Journal of Horticultural Science* **7**:93–97.

**Mayer D, Lunden J. 1999a.** Field and laboratory tests of the effects of fipronil on adult female bees of *Apis mellifera*, Megachile rotundata, and *Nomia melanderi*. *Journal of Apicultural Research* **38**:191–197 DOI 10.1080/00218839.1999.11101009.

**Mayer D, Lunden J. 1999b.** Residual bee poisoning bioassay. In: *Integrated pest and pollinator investigations.* Prosser: Washington State University, 2–5.

**Mayer D, Lunden J, Husfloen M. 1991.** Residual bee poisoning bioassay. In: *Integrated pest and pollinator investigations.* Prosser: Washington State University, 1–2.

**Mayer D, Lunden J, Jasso M. 1996.** Residual bee poisoning bioassay. In: *Integrated pest and pollinator investigations.* Prosser: Washington State University, 1–4.

**Mayer D, Lunden J, Johansen C. 1985.** Small scale bee poisoning bioassay. In: *Bee research investigations.* Prosser: Washington State University, 2–5.

**Mayer D, Lunden J, Kovacs G. 1997.** Susceptibility of four bee species (Hymenoptera: Apoidea) to field weathered insecticide residues. *Journal of Entomological Society of British Columbia* **94**:27–30.

**Mayer D, Lunden J, Miliczky E. 1988.** Residual bee poisoning bioassay. In: *Integrated pest and pollinator investigations.* Prosser: Washington State University, 1–5.

**Mayer D, Lunden J, Rathbone L, Miliczky E, Johansen CA. 1987b.** Residual bee poisoning bioassay. In: *Bee research investigations.* Prosser: Washington State University, 1–5.

**Mayer D, Lunden L, Rathbone L, Johansen C. 1986.** Residual bee poisoning bioassays. In: *Bee research investigations.* Prosser: Washington State University, 2–6.

**Mayer D, Patten K, Macfarlane R, Shanks C. 1994.** Differences between susceptibility of four pollinator species (Hymenoptera:Apoidea) to field weathered insecticide residues. *Melanderia* **50**:.

**Mayes M, Thompson G, Husband B, Miles M. 2003.** Spinosad toxicity to pollinators and associated risk. *Reviews of Environmental Contamination and Toxicology* **179**:37–71.

**Pashte V, Patil C. 2017.** Evaluation of persistence of insecticide toxicity in honey bees (Apis mellifera L.). *Indian Journal of Biochemistry and Biophysics* **54**:150–155.

**Sanchez-Bayo F, Goka K. 2014.** Pesticide residues and bees –a risk assessment. *PLOS ONE* **9**:e94482 DOI 10.1371/journal.pone.0094482.

**Summit Agro and Harvanta.** 50SL INSECTICIDE. Durham: Summit Agro.

**Vinchesi A, Boyle N, Walsh D. 2013.** Studies on alkali bees and pollinator pesticide safety in Washington State. In: *Western Alfalfa Seed Conference, Las Vegas, NV*.

**Waller G, Estesen B, Buck N, Taylor K, Crowder L. 1988.** Residual life and toxicity to honey bees (Hymenoptera:Apidae) of selected pyrethroid formulations applied to cotton in arizona. *Journal of Economic Entomology* **81**:1022–1026 DOI 10.1093/jee/81.4.1022.

**Walsh D. 2010.** Insecticide Efficacy Trials 2008-2009. In: *Integrated pest management on alfalfa seed: a two-year report.* Las Vegas: Western Alfalfa Seed Growers Association, 2008–2009.

**Walsh D, Boydston R, O'Neal S. 2008.** 2005-2007 Alfalfa seed research report. Kennewick: Northwest Alfalfa Seed Growers Association.

**Walsh D, Wine E, Groenendale D, Vinchesi A, Boyle N. 2016.** Pest and pollinator management on alfalfa seed 2015..