# Peer review of "Systematic review of residual toxicity studies of pesticides to bees and veracity of guidance on pesticide labels"

_PeerJ, doi:10.7717/peerj.16672_

## Round 0.1 · original submission · Major Revisions

The paper has some merits but needs revisions before the next round of review. The language of the paper needs attention. Please incorporate all comments raised by all reviewers and re-submit the paper with a point-to-point rebuttal letter.

Reviewer 1 ·

Basic reporting

This is an important review study on Systematic review of residual toxicity studies of pesticides to bees and comparison to language on pesticide labels using data from studies and the Environmental Protection Agency. Quality of review can be further improved by addressing following changes.
Abstract: Very long abstract. Write at least two to three lines why this review is important to conduct? Usually review is done on what is missed in literature, which you proved from existing published data. It is appreciated. Results and conclusion is written in relevant parts. No need here. Write your observation based on those result in the Abstract. Not just the re-writing the results in the Abstract
Introduction: Take some updated studies of 2022 and 2023 and enrich your your review. You used very old studies. Why it is important to pesticide application in the evening why not in the morning? is there any empirical/practical or even theoretical justification. Shed a light on it. See lines 55-65.
Write a paragraph at the end of introduction why this review is important and what is left in the literature. Write that gap.

Experimental design

Authors mentioned "We narrowed these studies down to 50 papers by excluding papers that met
153 any of the following criteria: (1) the study was not a primary source of data (e.g., review paper);
154 (2) bees were exposed to the pesticide applied to paper instead of on plant material; and/or, (3)
155 the study focused on residual toxicity of pesticides applied to plots but did not involve harvesting
156 plant tissue for caged bees." Why not primary data that have up-to-date information?

Validity of the findings

Accept my appreciation for writing good results and conclusion

Reviewer 2 ·

Basic reporting

The manuscript is clear and well-written, except for a few sections in the Discussion, as noted in the "Additional Comments" below.

The PRISMA flow diagram is incomplete -- some n values are missing and some don't add up. The PRISMA checklist appears to be the template and a completed checklist with responses or text from the manuscript is not included.

Experimental design

The authors present the results of their effort to compile a comprehensive list of RT25 values, indicating the residual toxicity of pesticides, to three species of bees.  They go on to perform a systematic comparison of these values with the language regarding bee toxicity on the pesticide label. All data collection and re-analysis is clear, with data provided in the supplemental materials.

Validity of the findings

The RT25 determinations and their comparison with label language is valid. The only discrepancy is a disagreement between statements in the results and discussion sections: On lines 369-370 there is the statement that "most studies deviated from the age of bees recommended by EPA", but this is not consistent with the statement in the Results on lines 278-280 where "only a quarter of studies using an older age of bees than recommended by EPA".  Which statement is correct?

Additional comments

The authors present the results of their effort to compile a comprehensive list of RT25 values, indicating the residual toxicity of pesticides, to three species of bees.  They go on to perform a systematic comparison of these values with the language regarding bee toxicity on the pesticide label.  While they report a fair amount of alignment, the discrepancies stymie efforts by pesticide applicators to reduce pesticide impacts on bees through appropriate timing of insecticide applications.  The manuscript does an excellent job discussing the nuances of the procedures for generation RT25s and how more effort is needed in determining the impact of deviations from EPA recommended methods.  Overall, this is a valuable contribution to efforts to improve regulatory toxicology and pesticide labeling to better protect bees.  Table 2 is a fantastic resource for researchers, educators and pesticide applicators. 

My main criticism of the manuscript, however, is with Table 2.  Why aren't the product names included here?  EPA includes product names on their RT25 web page.  The authors acknowledge in lines 429-430 that the inert ingredients in particular formulations may be important in determining toxicity, but for some reason (space possibly?) the formulation names are not included in Table 2.  Inclusion of product names tested would make this valuable resource even more useful. 

My other major comment is that the issue of application rate needs to be introduced at the beginning and mentioned in the methods/results.  As currently written, application rate is not mentioned until the Discussion.

Finally, there are parts of the discussion that could be rewritten -- I highlight specific sections in the line-by-line comments that follow.

The PRISMA flow diagram is incomplete -- some n values are missing and some don't add up. The PRISMA checklist appears to be the template and a completed checklist with responses or text from the manuscript is not included.

Abstract: 

23: specify that residual toxicity studies are performed using formulated pesticides

47: change "study" to "studies"

Introduction:

59: remove "their" in "their mortality"

78: remove "falls"

Materials and Methods: 

Include which authors conducted the search strategy in compliance with PeerJ "systematic review" guidelines 

148: Insert "a": We also evaluated _a_ series of . . . 

169: Change "variance of EPA methodology" to "variance from EPA methodology"

192: Why was it assumed that there were a total of 100 bees per cage when EPA guidance, which many studies follow, recommend 25?  Not sure if this makes any difference in the results?  Either clarify or remove.

Results:

273: remove "used" before "cotton

Discussion:

330-335: Application rate is mentioned here in the Discussion, but it isn't clear how application rate was incorporated into the results that are presented or how consideration of the application rate caused some studies to be excluded.  More information on application rate is needed in the Methods/Results.

336-338: The statement "which illustrates the extent to which researchers have not kept pace with the rate of pesticide product development" implies that it is the responsibility of independent researchers to conduct a this standardized test needed for registration.  It really should be up to USEPA and the registrants to communicate this information through the EPA's RT25 database.  Unless these tests aren't being required of registrants?  I think the authors should spread the blame around for the lack of publicly available RT25 values.

363-369: These sentences could be rewritten to clarify the potential effects of the number of bees on the outcome of the assay. The statement that non-Apis bees, for which fewer were placed in cages, may have lead to less contact with pesticide is not substantiated and are not consistent with the "concentrating/diluting" and "shorter/longer" statements in the previous sentence.  It really isn't clear to me how the number of bees may affect the outcome of these experiments -- work is needed in this area, as the authors stated earlier.  Maybe remove these sentences entirely?  Or rewrite to state that it isn't well-understood and more work is needed?

369-370: The statement that "most studies deviated from the age of bees recommended by EPA" is not consistent with the statement in the Results on lines 278-280 where "only a quarter of studies using an older age of bees than recommended by EPA".  Which statement is correct? 

401-404: These sentences could be rewritten to be more clear about the relationship between LD50 and RT25

405: Cyhalothrin is a pyrethroid, not an organophosphate

Acknowledgements:

Matthew Bucy is already listed as a co-author and should not be thanked here.

Table 2:

The heading "Greater than or less than 8 hours" seems out of place.

I'd really like to see the product name or some way to reference the actual formulated product tested.  While this might make the table unwieldy, it would also allow the table to be used as a practical reference.

Table 3:

This table is not referenced in the text and could be removed since mode of action is not discussed (and doesn't really need to be).

Figure 3: Is this figure needed?  This result is worth highlighting, but I'm not sure a figure to illustrate it is really justified.

Figure 4: This information might be better represented using a Venn diagram?

Figure 5: This figure should include information on products that don't align and have either shorter or longer RT25s compared to the label language.

·

Basic reporting

Overall the manuscript is well organized and written. The nature of the topic makes the language at times a bit confusing/convoluted, and I have suggested some places where the clarity could be improved (see notes on annotated PDF). I also think the Methods and Results might benefit from more creative use of flowcharts/tables to communicate the relationships between RT25 values, EPA guidance, and observed language on labels. The authors do a nice job situating their work relative to the literature and orienting the reader to the regulatory history and landscape with respect to bee residual toxicity guidance on pesticide labels (many readers are likely unfamiliar with these details). One exception is the discussion of cross-species differences in toxicity, which could be improved by incorporating more recent literature on this subject (linked in the comments). The results and discussion align well to the stated goals of the study.

The English is generally sound but there are some minor grammatical errors throughout (e.g. subject-verb agreement, tense agreement). I noted a few of these in the PDF and recommend that the authors review the entire manuscript carefully with this in mind. Also check that "studies", "papers", and "trials" are defined and used consistently - this is a key issue in data synthesis since often a single paper reports results from multiple experiments.

Table 3 does not appear to be cited in the text.

Figure 2 is showing very interesting results, but I think could be formatted more intuitively. For example it could be a stacked bar graph where positive values represent "longer" and negative values represent "shorter". Perhaps also consider putting species on the X axis and color-coding by formulation? What do the error bars represent here? (not explained in the caption, which should also include the total number of labels in the study for context - otherwise the # of occurrences is not very meaningful)

Figure 5 - I suggest splitting "don't align" into the two directions of mismatching (overly protective vs. overly permissive) since they have different implications.

Experimental design

The study begins with a systematic review of residual toxicity values toward bees, and then uses those values in combination with EPA values and guidance to evaluate whether pesticide labels provide accurate information to pesticide applicators. The overall approach appears sound. While the authors call the study a "meta-analysis" I'm not sure that is totally accurate; it does not appear that values from separate studies were statistically combined to derive an overall value (see Koricheva et al. 2013 Handbook of Meta-analysis in Ecology & Evolution). So "systematic review" may be more accurate.

The authors helpfully submitted a PRISMA diagram and checklist to document their data synthesis effort. The PRISMA diagram seems to be missing the sample sizes (n) in a few of the boxes. In the text, I found it a bit challenging to follow the sample sizes throughout. It could be helpful to summarize the # of observations (combos of AI/formulation/rate) as well as the # of papers/studies, since a given paper likely contributed multiple observations. And to ensure all figure/table captions report sample sizes (e.g. # of labels reviewed, # of pesticide comparisons).

There are a few aspects of the data synthesis that are a bit unclear to me. When multiple studies reported RT values for the same pesticide (e.g. acephate WP at 1 lb AI/ac), how were those values reconciled in performing the check against label language? In that example, the two provided values (7 hours and > 72 hours) lead to different conclusions. I'm also unclear how the language was checked against RT25 values when there were multiple rates reported that led to different conclusions. Finally, it would help put the results in context to know if EPA requires product-specific or formulation-specific RT25 data, or not. This could be explained early in the paper because it is relevant in a number of places.

A minor thing - I'm left unsure why RT25 values were re-calculated from the raw data (vs. using estimates from the original study) and curious to hear an explanation for that decision.

Validity of the findings

The relevant data and R code are shared, however it does not appear that the analyses can be reproduced from the submitted materials (that is, the submitted data file is not the input file to the code). Ideally it should be possible to run the code on the submitted data, but if that is not feasible I would recommend that the authors a) submit an example data file that can be run with the code (e.g. "Nomia Johansen et al. 1975 imdacloprid.csv", and b) further annotate the code to explain how the submitted data file relates to the dataset(s) on which the analyses were actually run.

The final database of RT25 values in Table 2 is an important contribution to the field, and I suggest that it be included in the supplemental material as a .csv file to enable re-use by other researchers (in addition to remaining in the text as a table).

The conclusions are clearly supported by the Results, and the interpretation in the Discussion is thoughtful. The Discussion could be made even stronger by further exploring the implications of differences in RT25 across species and the prevalence of overly permissive versus overly protective label language (to pest management and pollinator protection).

Additional comments

This is a nice study that makes an important contribution to improving bee protection. Despite the fact that "labels are the law" and a key way to communicate to pesticide applicators how best to protect bees from pesticide exposure, it is rare to see this kind of detailed evaluation of how labels are working in practice and whether they are aligned with research findings. I commend the authors for their creative work in this area and appreciate the RT25 database, which clearly was a big effort to assemble and is a service to the field.

---

## Round 0.2 · accepted · Accept

The paper is improved after revisions by incorporating all comments suggested by reviewers and is accepted for publication.